# Digital Single-Image Smartphone Assessment of Total Body Fat and Abdominal Fat Using Machine Learning

**DOI:** 10.3390/s22218365

**Published:** 2022-10-31

**Authors:** Gian Luca Farina, Carmine Orlandi, Henry Lukaski, Lexa Nescolarde

**Affiliations:** 1Medical Center Eubion, 00135 Rome, Italy; 2Medical Faculty, Tor Vergata University, 00133 Rome, Italy; 3Department of Kinesiology and Public Health Education, University of North Dakota, Grand Forks, ND 58202, USA; 4Department of Electronic Engineering, Universitat Politècnica de Catalunya, 08034 Barcelona, Spain

**Keywords:** two-dimensional digital imaging, machine learning, smartphone camera, fat mass, abdominal fat mass

## Abstract

Background: Obesity is chronic health problem. Screening for the obesity phenotype is limited by the availability of practical methods. Methods: We determined the reproducibility and accuracy of an automated machine-learning method using smartphone camera-enabled capture and analysis of single, two-dimensional (2D) standing lateral digital images to estimate fat mass (FM) compared to dual X-ray absorptiometry (DXA) in females and males. We also report the first model to predict abdominal FM using 2D digital images. Results: Gender-specific 2D estimates of FM were significantly correlated (*p* < 0.001) with DXA FM values and not different (*p* > 0.05). Reproducibility of FM estimates was very high (R^2^ = 0.99) with high concordance (R^2^ = 0.99) and low absolute pure error (0.114 to 0.116 kg) and percent error (1.3 and 3%). Bland–Altman plots revealed no proportional bias with limits of agreement of 4.9 to −4.3 kg and 3.9 to −4.9 kg for females and males, respectively. A novel 2D model to estimate abdominal (lumbar 2–5) FM produced high correlations (R^2^ = 0.99) and concordance (R^2^ = 0.99) compared to DXA abdominal FM values. Conclusions: A smartphone camera trained with machine learning and automated processing of 2D lateral standing digital images is an objective and valid method to estimate FM and, with proof of concept, to determine abdominal FM. It can facilitate practical identification of the obesity phenotype in adults.

## 1. Introduction

Obesity is a chronic multisystem disease that increases the risk of long-term medical complications, reduces life span, and increases health care costs. The World Health Organization estimated more than 650 million adults were obese in 2016, a value representing a three-fold worldwide increase since 1975 [1]. Obesity was deemed responsible for 4 million deaths in 2015, with two-thirds attributable to cardiovascular disease (CVD) [2]. The global economic burden of obesity is projected to be USD 1.2 trillion in 2025 [3]. Thus, identification of individuals with obesity is critically needed to reduce morbidity and mortality.

Methods to identify the obesity phenotype, designated as adiposity or excess body fat and abdominal fat accumulation, have limitations. Clinicians and epidemiologists classify body weight based on ranges of body mass index (BMI; weight/height^2^, kg/m^2^) to stratify obesity-related health risks in a population. Although insensitive to inter-individual differences in fat and muscle [4], BMI is a convenient surrogate for adiposity-associated risk factors related to metabolic perturbations leading to type 2 diabetes mellitus (T2D), CVD, and some cancers [5,6,7,8,9,10,11]. Dual X-ray absorptiometry (DXA) is a laboratory-based method that provides accurate estimates of adiposity but is limited in availability. Anthropometry (skinfold thicknesses and girths) and bioelectrical impedance are less costly, more time-efficient, and easier to perform methods but are less reliable and variable in accuracy [12,13,14].

Body fat distribution is another component of the obesity phenotype [5,6]. Initial observations identified that body shape, characterized using waist and hip circumference measurements, and specifically body fat in the upper compared to the lower body, was associated with obesity-related comorbidities and increased risk of mortality [15]. Later findings emphasized the importance of waist circumference (WC), a surrogate for visceral adipose tissue (VAT) [16], as a predictor of increased risk of T2D, CVD, and death. Anthropometric measurements at different anatomical landmarks of the abdomen are commonly used for routine estimation of WC with variable results [17,18]. Direct measurement of VAT is limited and requires medical imaging equipment (computed tomography (CT) or magnetic resonance imaging (MRI)), whereas DXA indirectly and inconsistently approximates VAT [5,18].

In contrast, digital imaging is a safe and rapid method to assess body shape and composition from photographic images [19,20]. Initial, and some contemporary, digital imaging methods use three-dimensional (3D) techniques with general access constrained by size and cumbersome equipment, immobility, and cost [21,22,23]. The current widespread public availability of smartphone cameras enables acquisition of two-dimensional (2D) digital image photographs that overcome the practical limitations of 3D digital imaging systems [19,24,25,26]. Contemporary use of digital imaging methods focuses solely on estimation of total body adiposity but offers an opportunity to assess localized fat in body regions such as the abdomen.

Computational machine learning is a mathematical tool increasingly used in medical research to develop algorithms designed to make accurate predictions from large data sets [27]. The application of machine learning to digital imaging analysis to estimate body fat is limited, with reports of poor reproducibility of body composition estimates, weak concordance with reference values, and proportional bias [24,25,26].

The purpose of this study was to validate an innovative DXA-trained, 2D digital imaging model derived from a smartphone and machine learning to predict body fat mass (FM) and to derive a model of 2D digital photography (DP) with machine learning to estimate abdominal FM in adults.

## 2. Materials and Methods

We recruited healthy adult Caucasian women and men aged 19 to 64 y by using advertisements and word of mouth to participate in this study, which was conducted at the Eubion Medical Center and Tor Vergata University, in Rome, Italy. Each prospective participant underwent a clinical examination and completed a health questionnaire to establish the absence of an unhealthy condition prior to participation. This study was approved by the Institutional Review Board of the University of Tor Vergata. Each participant provided written informed consent prior to participation in any testing.

### 2.1. Digital Photography

The patented operational principle of 2D DP is that the real value in metrics of each picture composing the surface of any digitally acquired image encased within a background can be computed by its occupation ratio within a digitally constructed virtual frame [28]. The acquisition and analysis of the 2D digital image follows a series of orderly steps (Figure 1). The operator first downloads and installs the Fit.Your.Outfit APP that is available in iOS and Android from the APP store. This APP ensures the adequacy of the technical quality of the photograph by using built-in sensors of modern smart devices and native libraries to detect specific anatomic nodal points (e.g., eye, nose, mouth, shoulder, elbow, hand, hip, knee, ankle, etc.), to identify parallaxes, improper distance from an individual to a camera, camera preventing small sized pictures that would yield coarse pixel size of silhouette, and to recognize incorrect arm extension and position, improper alignment of legs and feet, and non-horizonal head position. This quality control protocol provides visual warning instructions to the operator to ensure acquisition of high-quality digital images for analysis.

The operator uploads the original picture to the Cloud-based neural network educated by deep learning machine technology for automated intelligent conditioning with Python programming language, which removes the background using FasterRCNN (https://arxiv.org/abs/1506.01497) and a proprietary pre-trained model. This first step detects and extracts the human profile within the picture conditioned as a single, homogenous, white pixel (256-256-256) bitmap silhouette and black background (0,0,0) pixels. The percent of occupation of white compared to black pixels is utilized to compute a real pixel size. The original picture undergoes conditioning in a few seconds and is no longer available from the Cloud memory thus ensuring the anonymity of the individual. Detection begins with identification of landmarks at the eyes and the malleolus using Posenet (https://arxiv.org/abs/1803.08225). To construct the virtual frame, the pixel counts between lines set over the eyes and the malleolus form the frame’s vertical sides (y-axis), and the horizontal upper and lower limits (x-axis) are virtualized by a preset standard fraction of the vertical boundaries. The percent of occupation of white vs. black pixels is utilized to compute the real pixel (total white pixel count/total black pixel count) without dependence on distance and camera resolution. The detailed body detection and pixel size determination are achieved with DeepLabV3+ (https://arxiv.org/abs/1706.05587) and Cross-Domain Complementary Learning Using Pose for Multi-Person Part Segmentation (https://arxiv.org/abs/1907.05193). Proprietary equations are used to compute body and abdominal FM. Time from acquisition of digital image (photograph) to return of body composition estimate is approximately 12 s on a high speed WIFI connection.

### 2.2. Procedure

Each volunteer wore minimal, shape-conforming clothing and stood with a lateral side toward the camera without regard to background. The individual stood upright with their head positioned in the horizontal plane, arms fully extended alongside the body with feet and legs touching, and aligned sagittal to the camera to provide a lateral profile of the body (Figure 2). The smartphone cameras, either iOS- or Android-controlled, were of various resolutions, all above 50 megapixels. The software automatically scaled all the digital pictures to a single homogeneous resolution of 5 megapixels. They were either pre-positioned on a stable tripod or held by a second individual who directed the handheld smartphone camera with the lens pointed at the middle of the standing height of each study participant. The distance from the camera to the individual was 1.8 to 2.1 m (6 to 7 ft). Illumination in the room was not controlled.

### 2.3. Validation of a Prediction Model for Total Body and Abdominal FM

One hundred fifty-eight healthy adults (58% female) had standing lateral whole-body digital images taken with either iOS Apple smartphone (operating system 10 or newer) or Android (operating system 5 or more recent) with 5 megapixel minimum camera resolution. This relatively low resolution for contemporary higher resolution smart device cameras enables this method to be broadly usable.

Volunteers came to the laboratory in the morning after an overnight fast wearing form-fitting clothing. Standing height and body weight were determined using standard medical equipment (SECA Stadiometer and SECA 762 scale). Reference whole-body composition was determined with a GE Lunar iDXAncore sn 200278 using software version 14.10.022. In an independent sample of 63 adults (52% female), an experienced DXA technician identified lumbar vertebrae 2 through 5 (L2-5) in whole-body DXA scans and determined FM with appropriate software All participants were positioned supine and scanned within the dimensions of the DXA table.

### 2.4. Statistical Methods

Statistical analyses were performed using SYSTAT version 13 (Systat Corporation; San Jose, CA, USA) and version 19.0.3 (MedCalc Software Bvba, Ostend, Belgium). Descriptive data are expressed as mean ± SD. Statistical significance was set at *p* < 0.05.

We determined the reproducibility of conditioning of digital images from consecutive digital photographs of individuals, who were repositioned between acquisition of digital photographs, transformed into silhouettes, and then computed to the two FM values for each subject. Precision or technical error measurement (TEM) is reported as the average values (±SD), coefficient of variation, and mean differences (±SD).

Total body FM was estimated with a proprietary prediction model trained with DXA, exclusive of body weight as an independent variable, derived in an independent sample, and compared with DXA-determined reference FM measurements in groups stratified by gender. Strength of agreement between estimated and reference FM and abdominal FM was determined with the Linn concordance correlation coefficient [29]. Absolute pure error (APE) and percent error were used to describe model accuracy relative to DXA. The null hypothesis to test the linearity of the relationship between measured and predicted values was slope similar to 1 and intercept not different than 0. Measured and predicted FM values in each gender group were compared separately with a paired *t*-test. Bland–Altman plot was used to determine bias and limits of agreement (LOA) for the derived prediction model.

## 3. Results

### 3.1. Estimation of Whole-Body FM: Model Validation

Table 1 describes the physical characteristics of the participants in the validation study and shows the wide ranges of BMI and adiposity in the groups.

Another sample of adults participated in the reproducibility trial (females: *n* = 12; males: *n* = 15). Estimated FM was similar for the consecutive tests for females (22.59 ± 8.51 vs. 22.87 ± 8.52 kg, respectively; *p* = 0.83) and males (16.45 ± 7.06 vs. 16.31 ± 6.83 kg, respectively; *p* = 0.83) with mean differences between tests not different (*p* = 0.82) than 0 (females = 0.04 ± 0.04 kg; males = 0.03 ± 0.02 kg). Coefficients of determination and concordance coefficients were high (*p* < 0.0001) for each group (0.996 and 0.997, respectively).

The DP-predicted and DXA-determined FM values were similar in the female (22.6 ± 10.6 and 22.8 ± 10.9 kg, respectively; *p* = 0.28) and male (17.1 ± 7.5 and 17.5 ± 7.2 kg, respectively; *p* = 0.26) groups. Within each gender group, predicted and measured FM values were significantly correlated (R^2^ = 0.95 and 0.91; *p* < 0.0001) with concordance correlation coefficients of 0.95 and 0.91 for females and males (*p* < 0.0001), respectively. Absolute pure error (0.002 and 0.057 kg) and percent error (1.3 and 3%) were low in the female and male groups. The predicted and measured FM values were distributed linearly in the male and female samples with slopes similar to 1.0 and intercepts not different than 0 (Figure 3).

The distribution of the differences between measured and predicted FM values as a function of increasing FM revealed no significant trends (Figure 4). The bias or differences between predicted and measured FM values were not different than 0 (0.28 kg, *p* = 0.28; and −0.5 kg, *p* = 0.26) for females and males, respectively, with no significant (*p* = 0.2) trends or slopes relating difference values as a function of the average FM values. The limits of agreement (LOA; mean ± 1.96 SD) ranged from 4.9 to −4.3 kg and from 3.9 to −4.9 kg for females and males, respectively.

### 3.2. Estimation of Abdominal FM

An independent sample of healthy Caucasians adults participated in the pilot study to predict abdominal FM using 2D DP compared to DXA L2-5 scans (Table 2). Note the variability in total FM and abdominal FM.

Estimated abdominal FM values were similar for the consecutive tests with repositioning of each participant for females (1.99 ± 1.26 vs. 2.10 ± 1.28 kg, respectively; *p* = 0.83) and males (1.93 ± 1.25 vs. 1.91 ± 1.22 kg, respectively; *p* = 0.83) with mean differences between tests not different (*p* = 0.83) than 0 (females = 0.04 ± 0.04 kg; males = 0.01 ± 0.01 kg). Coefficients of determination and concordance coefficients were high (*p* < 0.0001) for each group (0.994 and 0.995, respectively).

The initial machine learning model provided abdominal FM estimates that were significantly (*p* < 0.0001) and linearly related to DXA abdominal FM values (R^2^ = 0.95 and 0.88) with concordance (R^2^ = 0.97 and 0.94) and slopes similar to 1 and intercepts not different than 0 (Figure 5).

## 4. Discussion

The present study demonstrates the high precision, concordance, and accuracy of a single, standing lateral 2D digital image analyzed using automated machine learning to estimate FM in adults with a wide range of adiposity. Reproducibility of 2D DP FM estimates was very high (R^2^ = 0.99) with high level of concordance between estimated and reference FM values (R^2^ = 0.91 to 0.95) and high precision (SEE = 2.3 to 2.4 kg). Absolute (0.002 to 0.057 kg) and percent errors (1.3 and 3%) between predicted and DXA-determined FM values were negligible. Predicted and reference FM values were similar (−0.3 kg, *p* = 0.28; and 0.5 kg, *p* = 0.26 for females and males, respectively), significantly correlated with high precision (R^2^ = 0.95 and 0.88; SEE = 2.4 and 2.3 kg for the female and male groups, respectively). Estimated and reference FM values were distributed similar to the line of identity. Examination of distribution of errors between paired 2D DP and DXA FM for individuals found minimal error or bias (0.5 and −0.3 kg) that was not affected by magnitude of FM (e.g., no proportional bias) with LOA of 4.9 to −4.3 kg and 3.9 to −4.9 kg for females and males, respectively. These findings indicate very high reproducibility and agreement between predicted and reference FM values with lower technical error and higher sensitivity than previous studies of 2D DP model development and validation.

The application of 2D digital imaging for body shape and composition assessment has evolved technically with variable results. Farina et al. [19] first developed and validated a multiple regression model to estimate FM based on gender-specific pixel occupancy derived using a commercial 2D smartphone camera from a single, standing lateral digital photograph of Caucasian adults wearing close-fitting, non-compressing apparel with legs together and arms extended along the mid-line of the body. This method used either Android version 4.2.2 on a Huawei G730 smart phone (resolution 540 × 960 pixels) or iOS 9.2 on an iPhone 5s (resolution 828 × 1792 pixels). It required the absence of inclusions in the background and operator-dependent delineation of the region of interest to delineate the photographic silhouette of an individual from the background. Inter-observer reliability was high (0.03% of pixel occupancy) with an average error of 0.02%. Gender-specific predicted FM and DXA-determined FM values were not statistically different, highly correlated (R^2^ = 0.982 and 0.991; SEE = 2.83 and 2.71 kg for female and males, respectively), without proportional bias, and LOA of 5.6 to −5.7 and 5.6 to −5.4 kg for females and males, respectively. Fedewa et al. [24] evaluated a proprietary 2D image processing system to estimate body volume of young Caucasian adults. Participants wore snug-fitting athletic clothing and stood facing a white photographic background with back to the camera, feet together, and arms abducted 45° from the torso in the coronal plane. A single digital image, including the head, arms, and feet, was obtained in the posterior view using a 64-g iPad Pro. Body volume estimated using digital image analysis was highly correlated with volume estimated from underwater weighing (R^2^ = 0.99, SEE = 0.68 L) with no difference between mean digital and densitometric body volume values (*p* = 0.58). Estimates of body fatness (%fat), calculated with body volume estimates derived from 2D DP and UWW using a three-component body composition model, were significantly correlated (R^2^ = 0.93) with LOA of 4.13 to −3.98%. Similarly, FM values were highly correlated (R^2^ = 0.93) with LOA from 2.96 to −2.76 kg. Nana et al. [25] captured 2D smartphone digital images of Asian adults in two poses: a frontal position with arms extended from the body, feet together and legs separated at shoulder width, and a right lateral position with feet together and arms extended along the torso and thigh in front of a green photographic screen. Twenty-four replicate images were obtained for each pose (three sets of eight consecutive digital photographs). Photo-images were analyzed with proprietary artificial intelligence-based software and yielded a participant-specific body contour and %fat estimates after input of gender, height and body weight. Smartphone-predicted %fat estimates had high concordance with DXA %fat measurements (R^2^ = 0.81 and RMSE = 2.8 and 2.9% for females and males, respectively) that were similar to DXA-determined %fat values. Bland–Altman plots revealed proportional bias with LOA of 5.6 to −5.8% and 5.6 to −5.4% for females and males, respectively.

Majmudar et al. [26] evaluated the use of convolutional neural networking, a type of machine learning, for analysis of smartphone 2D digital images to estimate %fat from two standing body positions. Adult participants wore form-fitting clothing without jewelry, socks, or shoes and were photographed in frontal and back poses with arms extended from the torso and legs separated. A smartphone-enabled camera was located 4 to 6 ft from the individual and positioned at knee height. A proprietary model to predict %fat derived in an independent sample was evaluated in this diverse sample of adults. Reproducibility of repeat estimates of %fat was high (R^2^ = 0.99) with an underestimation of 0.6 %fat between replicate determinations and a technical error (LOA) of 1.51 to −1.64 %fat. Percent body fat values derived from 2D DP and DXA were correlated in groups of female and male participants (R^2^ = 0.88). Linearity between measured and estimated fatness was not tested but a deviation of the intercept (2.92) from the line of identity (intercept same as 0) in the plot of the female data was reported. The mean absolute error between predicted and measured %fat was 2.16 ± 1.5% for the group and was greater for females than males (2.34 ± 1.14 vs. 1.88 ± 1.32%). The 2D image analysis program slightly underpredicted %fat. Bland–Altman plots revealed no proportional bias (−0.17) among males with LOA of 4.32 to −4.56% fat but significant proportional bias (−0.58; *p* = 0.002) and wider limits of agreement 4.9 to −6.1% in the female group. For the combined sample of females and males, bias (−0.42) approached significance (*p* = 0.062) with LOA of 4.7 to −5.55 % fat.

The burgeoning interest in the use of smartphone camera-enabled 2D DP reveals some important observations. Use of artificial intelligence, and specifically machine learning, for image analysis yields inconsistent validation results. Regardless of approach, 2D DP methods have relatively good reproducibility and similarities in group comparisons of estimated and predicted body fat variables. Although the investigators reported no significance between mean predicted and DXA-determined %fat values, proportional bias (e.g., tendency for greater errors from 2D DP with increasing %fat values) emerged and limits general use of these models [25,26]. In contrast, the results of the present study found no proportional bias between 2D DP-predicted and DXA-measured FM. Differences in test protocols, prediction models, and the quality control of image acquisition may contribute to measurement precision and prediction accuracy.

Use of proprietary prediction equations that fail to designate the physical characteristics and body shape variables used as independent variables confound the interpretation of findings of 2D DP measurement precision (reproducibility), trends in bias, and wide LOA [24,25,26]. A potential limitation is the failure to reveal the inclusion of body weight, a significant predictor of %fat and FM, hence affecting the independence of digitally acquired and derived body surface components in estimation of body shape and composition [24,25,26]. The present study, which is an evolution of the first use of 2D DP with a smartphone camera [19], uses machine learning to determine silhouette characteristics to derive a precise and accurate model to predict FM independently of body weight. The addition of artificial intelligence approaches offers increased objectivity in identification and quantification of measurements that resulted in reduced error and LOA in the present study compared to previous reports using 2D and 3D techniques [21,24,25,26].

Variation in body position for 2D DP photography is another influence moderating the reliable characterization of body shape. Adipose tissue distribution is gender-specific and generally indicates regional adipose accumulation in obesity [6]. Thus, digital images of lateral poses intuitively are more sensitive to identify and describe within and between individual differences in localized adipose depots compared to frontal or back positions. Evidence of the advantage of using different body positions, individually or in combination, compared to the lateral pose is lacking [24,25,26]. Use of multiple body poses also contributes to increased variability in estimating %fat (±1.5%) [26] compared to the increased precision (0.03 kg) of the lateral pose in estimating FM in the present study and reduced concordance between reference and 2D estimates of %fat [25] despite obtaining multiple consecutive digital photographs to overcome between-image variability in depicting body contour [25,26].

Our findings provide the first evidence that 2D DP is a valid method to estimate abdominal FM in adults. We found greater reliability (concordance coefficient = 0.992 to 0.995) and accuracy (0.01 to 0.04 kg; *p* = 0.83) than the more variable (R^2^ = 0.78 to 0.95) and less accurate (7 cm) 3D imaging systems for measurement of waist circumference [30,31]. Although preliminary, these results demonstrate a proof of concept for single, lateral 2D DP for future investigation.

The present study has some limitations. The composition of the sample is limited to Caucasian adults. To expand its applicability, follow-up studies should include a more ethnically heterogenous sample. It is also important to determine the effect of BMI, a traditional indicator of adiposity, on the validity of 2D DP machine learning FM and abdominal FM estimates. To date, all studies of 2D DP have been observational. Thus, it imperative to ascertain the validity and precision of the 2D DP method to estimate changes in whole body and abdominal composition in response to weight loss, physical activity, and metabolic disease. Growing use of artificial intelligence methods, such as machine learning, for algorithm development using digital imaging will benefit from inclusion of large data sets. Therefore, future studies should focus on large, diverse samples with broad representation based on race, physical characteristics, and interventions that alter body shape and composition.

In conclusion, the findings of the present study advance the use of single-image 2D digital imaging to estimate whole-body FM with the use of a novel machine learning APP that enables the very high reproducibility with negligible error in acquiring a single image for analysis. This APP coupled with an innovative machine learning model to estimate whole-body FM that yields values not different than reference DXA values. Importantly, the errors between individual 2D DP-estimated and DXA reference FM values are not negatively affected by the amount of FM. A second key result is the proof of concept that this quality control APP for image acquisition, together with a separate machine learning-derived model, accurately predicts abdominal FM, which has not previously been reported. These findings provide further incentive to adopt single scan lateral 2D DP with machine learning for practical assessment of FM in non-laboratory environments. They fulfill the need for a practical and feasible method to combine whole-body and regional body composition assessment for reliable cost-effective screening of individuals for risk of cardiometabolic disease [13,14,15,17] without the technical error of the manual measurement of regional girths [32,33] and afford future machine learning opportunities for relating body shape and composition to other clinical risk factors for chronic disease [34], without the limitations of 3D digital imaging systems.

## Figures and Tables

**Figure 1 sensors-22-08365-f001:**
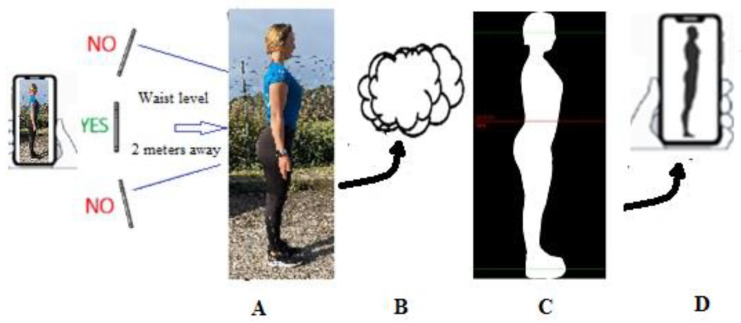
Illustration of the steps in the method of single-image, two-dimensional digital photography to characterize body shape and assess body composition: (**A**) image acquisition; (**B**) transfer to the Cloud for direct machine learning process including profile detection, extraction and conditioning of the image; (**C**) silhouette with body profile detect, extraction, conditioning; (**D**) return of silhouette with body shape and composition estimates.

**Figure 2 sensors-22-08365-f002:**
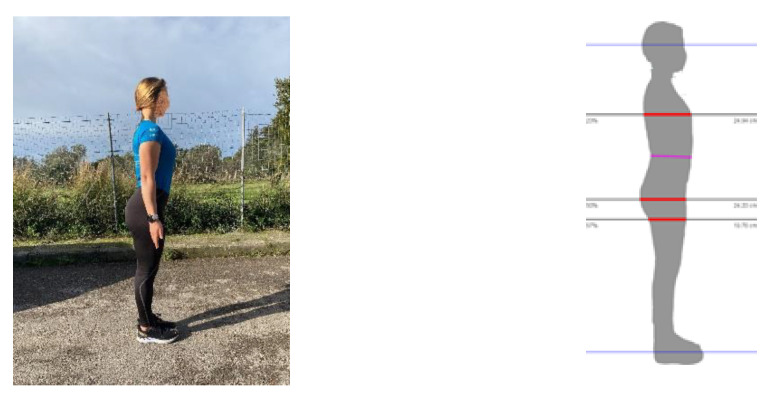
Images showing the sequence of digital image acquisition and software-determined landmarks for estimation of body fat. Digital photograph of lateral side of an individual in an unrestricted environment. Uploaded digital image of lateral surface of the individual with software-derived silhouette showing horizontal transverse lines designated with machine learning that identify the chest, belly, thigh, and calf sites.

**Figure 3 sensors-22-08365-f003:**
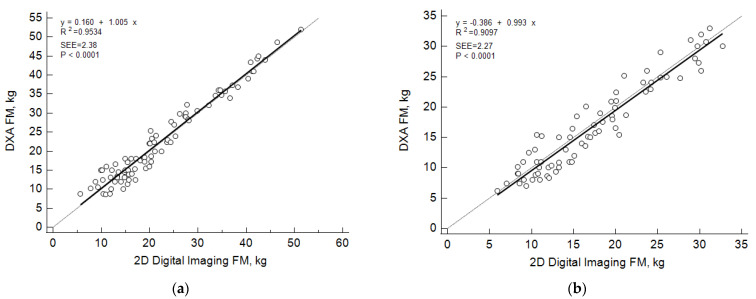
Linear regression plots of dual x-ray (DXA)-measured and two-dimensional (2D) digital imaging-estimated fat mass (FM) values in females (**a**) and males (**b**).

**Figure 4 sensors-22-08365-f004:**
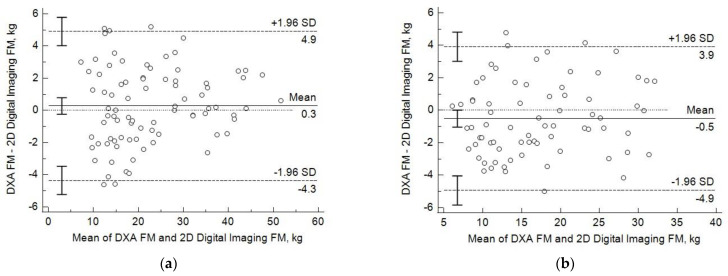
Bland–Altman plots illustrate the differences between individual dual X-ray absorptiometry (DXA)-measured and two dimensional (2D) digital image-predicted fat mass (FM) values as a function of the mean values of females (**a**) and males (**b**). Linear regression line describes the bias and 95% confidence intervals (1.96 SD) indicate the limits of agreement. Bars indicate 95% confidence intervals for mean and LOA values.

**Figure 5 sensors-22-08365-f005:**
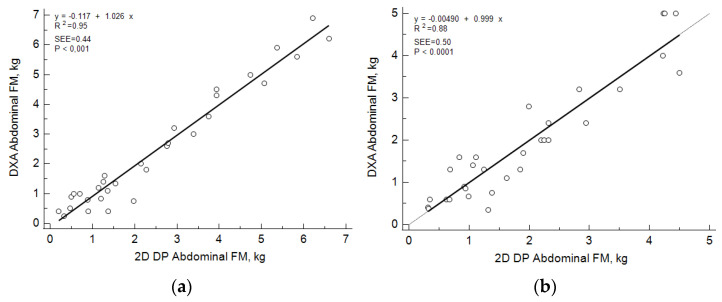
Linear regression plots of dual x-ray (DXA)-measured and two-dimensional digital imaging photography (DP)-estimated abdominal (L2–5) fat mass values in females (**a**) and males (**b**).

**Table 1 sensors-22-08365-t001:** Physical characteristics of participants. Values are mean ± SD (range of values).

*n*	Female	Males
*n*	84	74
Weight, kg	66.8 ± 13.1 (42.0 to 103.6)	80.1 ± 11.5 (63.0 to 109.0)
Height, cm	163.6 ± 6.7 (151.0 to 183.0)	178.8 ± 7.0 (163 to 199)
BMI ^a^, kg/m^2^	25.1 ± 5.1 (16.2 to 37.2)	25.1 ± 3.6 (19.1 to 37.1)
Fat mass ^b^, kg	22.8 ± 10.9 (8.6 to 59.2)	17.0 ± 7.5 (6.2 to 33)
Body fat ^b^, %	32.6 ± 10.1 (16.0 to 52.0)	20.8 ± 7.4 (9.5 to 37.2)
Fat-free mass ^b^, kg	44.5 ± 5.6 (31.0 to 58.7)	63.2 ± 8.5 (47.0 to 87.6)

^a^ Body mass index; ^b^ Dual X-ray absorptiometry.

**Table 2 sensors-22-08365-t002:** Physical characteristics of adult participants in model development to predict abdominal fat mass. Values are mean ± SD (range of values).

*n*	Female	Males
*n*	32	31
Weight, kg	67.0 ± 15.7 (42 to 103.0)	78.2 ± 12.2 (63 to 108.0)
Height, cm	162.5 ± 5.8 (152 to 174)	177.2 ± 7.4 (165 to 193)
BMI ^a^, kg/m^2^	25.5 ± 6.1 (16.2 to 37.2)	25.0 ± 4.1 (19.6 to 37.1)
Fat mass ^b^, kg	23.6 ± 12.8 (7.4 to 52.0)	17.0 ± 8.5 (6.2 to 35.0)
Abdominal fat mass ^b,c^, kg	2.45 ± 2.0 (0.24 to 6.9)	1.94 ± 1.41 (0.35 to 5.0)
Fat-free mass ^b^, kg	44.5 ± 5.6 (31.0 to 58.7)	63.2 ± 8.5 (47 to 87.6)

^a^ Body mass index; ^b^ dual X-ray absorptiometry; ^c^ lumbar vertebrae 2–5 area.

## Data Availability

Not applicable.

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
