# Peer review of "Digital Single-Image Smartphone Assessment of Total Body Fat and Abdominal Fat Using Machine Learning"

_sensors, 2022, doi:10.3390/s22218365_

Round 1
Reviewer 1 Report
The study is good, but the authors should address the following points.
1. Adding a new section to discuss relevant previous studies.
2. Indicate the most important main contributions in this study.
3. Add a figure to clearly explain the study methodology.
4. Tables 1 and 2 contain some characteristics acquired from the participants. The question is, are all the characteristics of the participants by means of pictures or some by a blood test?
5. The results should be explained more clearly and specify the accuracy, sensitivity and specificity.
6. Discussion about whether there are outliers or missing values and how they were addressed.
7. The Conclusion must be separated into a new section.

Reviewer 2 Report
The contribution of the paper is not clear, please rewrite it.
Reviewer 3 Report
The authors have proposed a method based on machine learning which employs smartphones to estimate body fat. The manuscript is written well and the topic is interesting. I think the level of the work, its novelty, and significance is acceptable and I suggest publishing it as it is.
Reviewer 4 Report
The main contribution and proposed approach have some novelty in contribution. Revision in terms of technical details is needed before publication. So, some comments are suggested to describe technical details.
1. Your proposed approach is performed on smartphones. So, it is suggested to discuss about the runtime of your proposed approach briefly.
2. Is the performance of your proposed approach sensitive to distance of human and camera? Discuss in a clear way
3. As described in section 2.2, a deep learner is used for image analysis. Discuss briefly about the structure of used deep learner. (layer types, number of layers, number of output neurons, etc)
4. The final output of your proposed approach can be used in medical applications such as immunology parameters. For example, I find a paper entitled “Study some immunological parameters for Salmonella Typhi patients in Hilla city”, which has relation. Cite this paper and discuss about it as one of the advantages of your proposed approach.
5. The quality of the figure 3 is low.
6. What is the relation between white pixels and wright?
Round 2
Reviewer 1 Report
Accept in present form
Reviewer 2 Report
I have no further comments.
Reviewer 4 Report
The revised version is better than original submission in terms of paper organization and technical details. Most of comments are considered in the revised text. The proposed method has enough novelty in methodology and contribution.